# Genome-Wide Identification of the CDPK Gene Family and Their Involvement in Taproot Cracking in Radish

**DOI:** 10.3390/ijms242015059

**Published:** 2023-10-11

**Authors:** Qian Yang, Yan Huang, Lei Cui, Caixia Gan, Zhengming Qiu, Chenghuan Yan, Xiaohui Deng

**Affiliations:** Key Laboratory of Vegetable Ecological Cultivation on Highland, Ministry of Agriculture and Rural Affairs, Hubei Key Laboratory of Vegetable Germplasm Innovation and Genetic Improvement, Institute of Economic Crops, Hubei Academy of Agricultural Sciences, Wuhan 430070, China; yangqian@hbaas.com (Q.Y.); huangyanzi@hbaas.com (Y.H.); cuilei18062099715@hbaas.com (L.C.); gancaixia@hbaas.com (C.G.); qiusunmoon@hbaas.ac.cn (Z.Q.)

**Keywords:** *Raphanus sativus*, taproot cracking, calcium-dependent protein kinase (CDPK), genome-wide analysis, transcriptomics

## Abstract

Taproot cracking, a severe and common physiological disorder, markedly reduces radish yield and commercial value. Calcium-dependent protein kinase (CDPK) plays a pivotal role in various plant developmental processes; however, its function in radish taproot cracking remains largely unknown. Here, 37 RsCDPK gene members were identified from the long-read radish genome “QZ-16”. Phylogenetic analysis revealed that the CDPK members in radish, tomato, and *Arabidopsis* were clustered into four groups. Additionally, synteny analysis identified 13 segmental duplication events in the RsCDPK genes. Analysis of paraffin-embedded sections showed that the density and arrangement of fleshy taproot cortex cells are important factors that affect radish cracking. Transcriptome sequencing of the fleshy taproot cortex revealed 5755 differentially expressed genes (DEGs) (3252 upregulated and 2503 downregulated) between non-cracking radish “HongYun” and cracking radish “505”. These DEGs were significantly enriched in plant hormone signal transduction, phenylpropanoid biosynthesis, and plant–pathogen interaction KEGG pathways. Furthermore, when comparing the 37 RsCDPK gene family members and RNA-seq DEGs, we identified six RsCDPK genes related to taproot cracking in radish. Soybean hairy root transformation experiments showed that RsCDPK21 significantly and positively regulates root length development. These findings provide valuable insights into the relationship between radish taproot cracking and RsCDPK gene function.

## 1. Introduction

Calcium (Ca^2+^) ions are vital nutrients and signals that play key roles in plant growth, development, and stress adaptation [1]. For example, Ca^2+^ is required to maintain the stability of cell walls and membranes [2,3] and is crucial for root hair development and root cortical cell elongation [4,5].

Several types of Ca^2+^ sensors exist in plants, including calmodulin (CaM), CaM-like protein (CaML), calcineurin B-like protein (CBL), calcium-dependent protein kinase (CDPK/CPKs), CDPK-related kinase (CRK), and CaM protein kinase (CCaMK) [6]. Among these, CaM, CaML, and CBL lack kinase domains [7,8]. Additionally, variations in CDPK, CRK, and CCaMK are likely caused by differences in the number and motif features of EF hands (the structural orientation of the two *α*-helices: E and F) [9]. CDPK contains kinase, auto-inhibitory, and regulatory domains that include four calcium-binding EF hands with conserved D-x-D motifs, whereas CRK contains only three calcium-binding EF hands in its regulatory domain [10]. Unlike CDPK, CCaMK contains Ser/Thr kinase and visinin-like EF-hand calcium-binding domains with three Ca^2+^ binding sites [11,12].

CDPKs are vital regulatory proteins that decode calcium signals triggered by various developmental and environmental stimuli and translate them into specific physiological responses [13]. Specifically, when plants are exposed to stress stimuli, the cellular concentration of Ca^2+^ undergoes distinct spatiotemporal changes, generating specific Ca^2+^ signals [14]. Subsequently, CDPKs are activated by the direct binding of Ca^2+^ to the CaM-like domain, enabling them to respond to these changes, regulate downstream targets, and induce various physiological processes in plants, including plant growth, development, and abiotic and biotic stress responses [15]. During this process, the EF hands of CDPK bind calcium ions and participate in gene regulation [16]. Moreover, CDPKs with various specific phosphorylation substrates mediate pathways and are considered as integrators of plant stress signaling and development [17].

Most CDPKs are involved in regulating plant growth, development, and stress responses [18]. For example, corn root protein kinase is expressed in root tissues and participates in Ca^2+^-dependent signal transduction [19,20]. As such, CDPK is considered as a vital signaling protein for root hair and cortical root cell elongation that directly or indirectly modulates cell wall development [21]. In *Arabidopsis*, *AtCDPKs* regulate auxin signal transduction via the phospholipase A pathway during root development [22]. In *Medicago truncatula*, reducing the expression of *CDPK1* alters the expression of root cell wall genes and increases lignin deposition in the cell wall [21]. CDPKs are also involved in adventitious root formation in cucumber [23]. In *Arabidopsis*, abscisic acid inhibits root growth by promoting ethylene biosynthesis [24]. Additionally, *AtCDPK16* mediates *GLR3.6* to regulate root growth [25]. 

Radish (*Raphanus sativus*, 2n = 18) belongs to the *Brassicaceae* family and is cultivated worldwide as an edible root vegetable. Unfortunately, radishes often suffer from untimely root cracking, resulting in decreased crop quality and root yield and economic losses [26]. Cracking development is a complex phenomenon involving multiple genes that may be related to cell wall metabolism, lignin biosynthesis, calcium transport and signaling, hormone metabolism, and water transport [27]. Cracking occurs due to the flexibility and elasticity of the epidermis and the presence of thinner and looser cell walls in the pericarp [28,29]. Several calcium-related genes exist in cracked fruits but not in non-cracked fruits [30]. Additionally, exogenously applied calcium can inhibit cell wall disassembly, promote cell wall strengthening, and prevent fruit cracking [31,32]. In oranges, foliar spraying with calcium nitrate decreases the expression of cell wall-degrading enzyme genes in the peel [33]. In nectarine, nano-Ca reduces fruit cracking by more than 20% [34]. Therefore, the genes involved in calcium binding, ion transport, and Ca^2+^ signal transduction play important roles in regulating cracking. In radish, two calcium regulatory genes (*RsANNAT* and *RsCDPK*) are positively associated with taproot cortex cracking [35]; however, the relationship between taproot cracking and *RsCDPK* genes remains poorly understood.

Although Ca^2+^ ions reportedly influence the cracking phenotype by regulating cell wall development, the role of CDPKs in taproot cracking has not been systematically investigated in radish. In this study, we aimed to identify RsCDPK genes and perform an evolution analysis of these genes using the assembled long-read radish genome “QZ-16“ [36]. Furthermore, we investigated the functions of RsCDPK genes in radish taproot cracking by analyzing differentially expressed RsCDPK genes and comparing the non-cracking radish variety “HongYun” with the cracking radish variety “505”. Ultimately, we identified six RsCDPK genes that may play a role in radish taproot cracking. Soybean hairy root transformation experiments confirmed that *RsCDPK21* positively regulates root length development. These findings improve our understanding of the taproot cracking mechanism and provide a theoretical basis for further investigation into the functions of CDPK genes in radish taproot development.

## 2. Results

### 2.1. Identification of RsCDPK Gene Members in Radish

A total of 37 putative RsCDPK genes were identified in the long-read radish genome “QZ-16” using the HMMER 3.0 software based on the P kinase domain (PF00069) and EF hand 7 (PF13499) from the Pfam database. All genes were verified using NCBI for Bath CD search. The RsCDPK genes were renamed based on their closest *Arabidopsis* homologs and chromosomal distributions. We analyzed the basic physicochemical properties of the RsCDPK members, including protein length (aa), isoelectric point (pI), and molecular weight (MW). The identified RsCDPK genes contained variable amino acid lengths, ranging from 354 to 634 aa, with an average length of 530.46 aa. The MW of the 37 RsCDPKs ranged from 39.78 to 70.18 kDa, with an average MW value of 59.56 kDa. pI values ranged from 5.05 to 9.35. Subcellular localization predictions showed that all RsCDPKs were presumptively located in the nucleus (Appendix A).

### 2.2. Chromosomal Position and Collinearity Analysis of RsCDPK

The chromosome location information of the 37 RsCDPK genes was visualized using TBtools software (v1.1041) (Figure 1). RsCDPK genes were dispersed on all *R. sativus* chromosomes. Approximately 68% of RsCDPK genes were predominantly located on Rs6 (7), Rs4 (7), Rs7 (6), and Rs5 (5), whereas only one RsCDPK gene was found on chromosomes 1 and 8. The number of CDPK genes identified in *Arabidopsis* (34) [9] and *Brassica rapa* (49) [37] indicates no correlation between changes in genome size and the number of CDPK gene family members [10].

The relationship between collinearity and default threshold values was clarified using the multiple collinearity scan toolkit (MCScan X) [38]. The results revealed 13 segmental duplications (Appendix A) and no tandem duplications of gene pairs (Figure 2). Segmental duplication events are recognized as an important mode for expanding gene family members [39].

### 2.3. Analysis of Gene Structure and Conserved Motifs of RsCDPK

To better explore the structural characteristics and evolutionary relationships of RsCDPK genes, RsCDPK protein sequences were used to construct a phylogenetic tree and analyze conserved motifs using MEGA 7 and MEME, respectively. According to the phylogenetic tree, the 37 RsCDPK members were categorized into four clades (Figure 3A). The number of motifs in the 37 RsCDPK genes ranged from 9 (RsCDPK37) to 14 (RsCDPK6, RsCDPK19, and RsCDPK5), with an average number of 12.4 (Figure 3B and Appendix A and Appendix A). Motifs 1, 2, 9, and 11 were distributed across all groups. Except for RsCDPK37, all genes contained motif 4. In addition, motif 14 was distributed only in group 4 and motif 15 in group 1. Group 4 contained two of motif 8. These results suggest that similar gene structures were divided into consistent clades.

Subsequently, we analyzed the gene structure and evolutionary relationships of the RsCDPK genes. The exon numbers of the 37 RsCDPK genes ranged from 4 (RsCDPK37) to 12 (RsCDPK32, RsCDPK18, RsCDPK16, and RsCDPK23), with an average number of 7.65. The number of exons in the last group was >11, with an average of 11.8. In addition, the number of exons in the top three genes (RsCDPK36, RsCDPK29, and RsCDPK13) with the longest DNA sequences was 8, 8, and 7, respectively (Figure 3C). This number variability indicates the evolutionary events of CDPK genes in terms of gene size changes [10].

### 2.4. Phylogenetic and Motif Analysis of CDPK Genes

To identify the evolutionary relationships between CDPK genes, a phylogenetic tree of 100 CDPK genes was constructed from radish (37), *Arabidopsis thaliana* (34), and tomato (29) plants using MEGA 7 based on the muscle method and neighbor-Joining tree (1000 bootstrap replicates). *A. thaliana* and tomato are model plants of *Cruciferae* and *Solanaceae* (fruit cracking occurs often), respectively, and were used to analyze and identify the evolutionary relationships of RsCDPK gene members. The results indicated that the CDPK members were divided into four groups (Figure 4). Groups 1–4 had 13, 6, 13, and 5 members in radish; 10, 13, 8, and 3 members in *Arabidopsis*; and 13, 8, 6, and 2 members in tomato, respectively. Group 1 had the largest number of CDPK members (36), whereas group 4 had the fewest (10). CDPK genes belonging to the same clade generally indicate homologous functions among members.

### 2.5. Morphological and Cytological Characterization of HongYun and 505

The high-generation inbred radish lines HongYun and 505 were tolerant and susceptible to root cracking, respectively. We observed no significant differences in the size of the leaves or fleshy roots at the mature stage (Appendix A). However, the fleshy taproot of 505 was cracked (Figure 5C), whereas that of HongYun showed normal development (Figure 5A). Cytological observations of the cortex and flesh of the fleshy taproot showed significant differences between the cortical cells of the non-cracking radish HongYun and those of the cracking radish 505 (Figure 5B,D and Appendix A). The cortical cells of HongYun (approximately 127 cells/50 μm vision) were characterized by a smaller size, larger number, and more compact arrangement than those of 505 (approximately 95 cells/50 μm vision) (Appendix A). In addition, we observed no significant differences in the size or number of taproot flesh cells (HongYun: 242 cells/200 μm vision and 505: 234 cells/200 μm vision, approximately) (Appendix A). These results indicate that the density and arrangement of fleshy taproot cortical cells are important factors that affect radish cracking.

### 2.6. Transcriptome Data Analysis

To explore the molecular mechanism of fleshy taproot cracking in HongYun and 505 at the mature stage, we sequenced the transcriptomes of three non-cracking gene pools (HongYun_1, HongYun_2, and HongYun_3) and three cracking gene pools (505_1, 505_2, and 505_3) in two radish varieties (HongYun and 505).

After preprocessing the raw reads, the number of high-quality reads in each of the six samples ranged from 34,204,982 to 46,469,346. A total of 34.40 Gb of clean bases were obtained, with each sample containing more than 5.13 Gb. The Q30 percentage was higher than 90.27%, indicating a high-quality data set. Additionally, the N% was less than 0.000803%, demonstrating a low level of sequencing errors (Appendix A). To assess the reproducibility of the transcriptional levels, Pearson’s correlation coefficients were calculated [40]. The coefficients among the three non-cracking replicates were equal to or greater than 0.91. Similarly, the coefficients among the three cracking replicates were ≥0.95 (Figure 6A). Based on these results, the RNA-seq data were deemed suitable for subsequent analysis.

The clean reads obtained from two pools were aligned to the assembled long-read reference genome of the radish cultivar “QZ-16” [36]. A significant proportion of the clean reads, specifically more than 78.86%, were successfully mapped to the “QZ-16” genome. Of these mapped reads, over 96.14% were uniquely mapped, while 3.53–3.86% were mapped to multiple locations on the genome (Appendix A). By analyzing the RNA-seq data from the six replicates, a total of 30,818 genes were identified. Subsequently, these genes were subjected to BLASTx searches against various databases, including GO, KEGG, Swiss-Prot, and NR. As a result, 30,436 genes were annotated, with varying levels of alignment to the databases. Specifically, 20,003 genes (64.91%), 12,078 genes (39.19%), 25,229 genes (81.86%), and 30,395 genes (98.63%) aligned to the GO, KEGG, Swiss-Prot, and NR databases, respectively. Moreover, 5755 differentially expressed genes (DEGs) (505/HongYun), comprising 3252 upregulated and 2,503 downregulated genes, were identified between the cracking and non-cracking fleshy taproot cortices (Figure 6B).

### 2.7. GO and KEGG Analyses of DEGs

According to the transcriptome data, we analyzed the root-cracking mechanism in 3252 genes in 505 (upregulated/cracking) and the root non-cracking mechanism in 2503 genes in HongYun (downregulated/non-cracking). Of the 3790 gene ontology (GO)-annotated DEGs (2166 upregulated and 1624 downregulated) in the RNA-seq data, 497, 46, and 190 GO categories were significantly enriched (*p* < 0.05) in biological processes (BPs), cellular components (CCs), and molecular functions (MFs), respectively, comprising 8164 functional groups (Appendix A). Among the top 10 BP categories, the response to oxygen-containing compounds (GO: 1901700) and wounding (GO: 0009611) may be involved in the plant stress response process after fleshy taproot cracking. The response to water deprivation (GO: 0009414) and water (GO: 0009415) indicated that water-related pathways are important factors for fleshy taproot cracking. Among the significantly enriched pathways in the CC category, only the anthranilate synthase complex pathway contained upregulated genes. In the MF category, secondary active transmembrane transporter activity (GO: 0015291), symporter activity (GO: 0015293), inorganic anion transmembrane transporter activity (GO: 0015103), and solute cation symporter activity (GO:0015294) were within the top 10 most significant terms. Some pathways related to membrane transport were also enriched.

A total of 1252 Kyoto Encyclopedia of Genes and Genomes (KEGG)-annotated DEGs (803 upregulated and 449 downregulated) were assigned to 125 KEGG pathways. These included cellular processes, environmental information processing, genetic information processing, metabolism, and organismal systems. Among them, plant hormone signal transduction, phenylpropanoid biosynthesis, and plant–pathogen interactions were significantly enriched (Appendix A). In the plant hormone signal transduction pathway, 71 genes were upregulated, and 33 genes were downregulated. Ethylene was only found in the upregulated genes (11 members: ERF/5, EIL/4, ERS2, and mPK6). In addition, auxin levels (GH3/8, IAA/8, and SAUR/5) were 1.75-fold those in crack-resistant radishes, whereas jasmonate levels were 8-fold those in crack-resistant radishes (maximum) (JAR/2, MYC/2, and TIFY/12). In the phenylpropanoid biosynthesis pathway, the top 3 of 56 upregulated genes were PER, IGMT, and BGL, which included 17, 9, and 15 genes, respectively. In the plant–pathogen interaction pathway, the most abundant gene family of the 62 upregulated genes was the calcium pathway (18 genes). These results indicate that hormone-related signaling pathways play an important role in regulating root traits.

### 2.8. CDPK Genes Related to Fleshy Taproot Cracking

To explore the regulatory role of CDPK family genes in the development of radish roots, we retrieved CDPK gene members in the DEGs from transcriptome data. Venn diagram analysis showed that six RsCDPK genes were present in both the 5755 DEGs and 37 RsCDPK members (Figure 7A). The expression patterns of the six RsCDPK genes are shown in a heatmap (Figure 7B). Notably, all these genes were upregulated in the cracking-susceptible variety. To validate these expression patterns, the levels of the six RsCDPK genes were assessed using RT-qPCR (Figure 7C, Appendix A). The RT-qPCR results corroborated the RNA-seq data, thereby confirming the accuracy of the transcriptome data.

### 2.9. RsCDPK21 Significantly and Positively Regulated Root Length in Radish

To verify the function of the six RsCDPK genes identified by Venn diagram analysis of the RsCDPK gene family members and transcriptome DEGs in regulating root development, the RsCDPK21 gene with the highest expression levels (FPKM values) was selected for further analysis. A soybean hairy root transformation experiment was performed to analyze the biological functions of the three identified genes and determine the relative expression levels of RsCDPK21 using an RT-qPCR assay (Figure 8A, Appendix A). The transgenic root lengths of the empty vector (EV) and RsCDPK21OE showed significant differences 14 days after rhizobial inoculation (Figure 8B). Among them, the average root length of RsCDPK21OE was substantially longer (169.1 mm) than that of the EV (99.6 mm) (Figure 8C, Appendix A). The RsCDPK21OE transgenic hairy root length showed a 68.76% increase when compared with that of EV. Cytological observations of the transgenic root of RsCDPK21OE (approximately 10.32%) showed that the proportions of xylem and phloem (root cross-sectional area) were higher than those of EV (approximately 32.97%) (Appendix A). These data indicate that RsCDPK21 is a positive regulator of taproot development and may regulate taproot cracking in radish.

## 3. Discussion

### 3.1. Taproot Cracking Formation in Radish

Cracking is a complex trait and a consequence of a combination of genetic, environmental, and biochemical factors, along with nutrient content [41]. For example, tomato fruit cracking is controlled by quantitative trait loci (QTL) × environmental interaction effects, indicating that genetic regulation is a key factor [42]. However, taproot cracking often occurs during the growth and ripening stages, significantly decreasing the yield and commercial value of radishes [43]. Unbalanced watering and water status are key factors that increase radish root cracking [44,45]. Excess water frequently causes cracking during the growth stage, leading to imbalances in cations such as Ca^2+^ [27]. In radishes, the taproot cortex plays an important role, acting as a mechanical protective barrier and defending against biotic and abiotic factors. However, cracking resistance differs among radish varieties, with the characteristics of the fleshy root cortex playing an important role. In this study, a high-density and compact arrangement of fleshy taproot cortex cells constituted important characteristics that reduced radish cracking (Figure 5). Clarifying the processes regulating radish root formation and development in terms of genetic analysis may enable breeders to develop new radish varieties that are tolerant to root cracking and provide a solution to taproot cracking in radish [35]. Some QTLs for head-splitting resistance have been detected and designed for molecular marker-assisted selection in cabbage [46,47]. The identification of genes involved in taproot cortex development, such as those related to the cell wall and cell density, provides new insights into the role of genetic factors in taproot cracking and the preferential breeding of new non-cracking radish varieties.

### 3.2. Cracking: A Complex Trait

Cracking is a quantitative trait involving a complex network of many genes, including those involved in cuticular membrane and cell wall metabolism, cutin biosynthesis and deposition, cuticular wax biosynthesis, water transport, calcium transport and signaling, starch and sucrose metabolism, and fruit hormone metabolism and transcription factors potentially involved in fruit cracking [27]. Therefore, cloning and identifying genes related to the regulation of cracking traits pose a considerable challenge. Some related genes have been evaluated and identified to be involved in the cracking process, including the wax composition-related gene CER6 [48] and the β-galactosidase gene TBG6 [49] in tomato, the expansin gene MdEXPA3 in apple [50], the expansin genes LcExp1 and LcExp2 in litchi [51], and the cell wall development related genes XET1, XET2, and DHCR24 in watermelons [52]. Moreover, calcium has an important role in affecting the cracking phenotype. CDPK functions are enhanced by Ca^2+^, implying that CDPKs are important for Ca^2+^/calmodulin-mediated signaling pathways [53]. Few studies have been conducted on the mechanism of calcium-related genes in cracking; however, several studies have revealed that exogenous calcium application can significantly reduce fruit cracking. For example, the downregulation of calcium transport and signaling genes such as CIPK, CML, and CNGC lead to cracking in litchi [54]. Meanwhile, the RsCDPK genes controlling radish root cracking have been identified by QTL mapping [35]. Identifying key calcium-related genes that regulate cracking is essential for radish breeding. Therefore, calcium signaling genes are an important and potential research direction for analyzing the cracking traits of radish fleshy roots.

### 3.3. CDPK Gene Members Related to Taproot Development and Cracking

CDPK is considered as a vital signaling protein for root hair and cortical root cell elongation, which directly or indirectly modulate cell wall development [21]. The fleshy taproot is an important economic and agronomic trait in radish. Hence, research on CDPK regulation of root development mechanisms has a potential application value for radish breeding. For example, AtCDPK16 mediates GLR3.6 to regulate root growth in *Arabidopsis* [25], and reducing the expression of CDPK1 alters the expression of root cell wall genes and increases lignin deposition in the cell wall in *M. truncatula* [21]. This indicates that CDPK has the potential to regulate the number of root cells and lignin content of cell walls, which is important theoretical support for cultivating crack-resistant radish varieties. This conclusion is consistent with the results of the root cytology of crack-resistant root materials in the present study (Figure 5B,D). Additionally, the RsCDPK21OE material showed a significant increase in root length (Figure 8B), confirming the important role of CDPK genes not only in biotic and abiotic stress but also in root development. These recent advances and findings of the current study provide a feasible scientific foundation for exploring the mechanism whereby CDPK genes regulate the fleshy taproot-cracking phenotype in radish. Furthermore, the CDPK gene family is expanded with high sequence conservation [16,55], indicating that the results of the present study can also provide a theoretical basis for fruit-cracking trait research.

Moreover, the hormone-related signaling pathways that play an important role in regulating root traits have been confirmed [27]. Recent research shows that most CDPKs are involved in regulating plant growth, development, and stress responses, especially phytohormone regulation. For example, NaCDPK4 and NaCDPK5 are involved in jasmonic acid biosynthesis in tobacco [56,57]. Additionally, compared with the pericarp of cracking-resistant cultivars, that of cracked lichi has a higher abscisic acid content, increased ethylene and jasmonic acid biosynthesis, and decreased auxin and brassinosteroid biosynthesis [58]. In jujubes, jasmonic acid biosynthesis is associated with fruit cracking [59]. Meanwhile, in the current study, ethylene, auxin, and jasmonate gene expression levels were upregulated in cracking radish when compared with those in crack-resistant radish. Therefore, CDPKs can regulate plant growth and development by mediating plant hormone signals and may play a crucial role in regulating taproot cracking.

## 4. Materials and Methods

### 4.1. Identification and Analysis of CDPK Family Genes in Radish

To accurately identify members of the CDPK family, the long-read radish genome “QZ-16” (ENA number: PRJEB37015) was used to screen for RsCDPK genes in radish. Initially, HMMER 3.0 software was employed to identify potential CDPKs by searching for Hidden Markov Model profiles (hmmsearch, E value ≤ 1 × 10^−5^) of the Pkinase domain (PF00069) and EF hand 7 (PF13499), which were obtained from the Pfam database (http://pfam.xfam.org, accessed on 7 August 2023). Subsequently, all putative CDPKs were re-tested using NCBI for Bath CD search (https://www.ncbi.nlm.nih.gov, accessed on 9 August 2023). Eventually, all members of the CDPK gene family were obtained and named based on their respective positions on the chromosomes in radish. The fundamental physicochemical properties of RsCDPK proteins, including the pI and MW, were analyzed using the ExPASy website (https://web.expasy.org, accessed on 20 August 2023).

### 4.2. Chromosomal Position and Collinearity Analysis of RsCDPK

The chromosomal distribution of the RsCDPK genes was visualized using TBtools software (v1.1041) [60] based on the list of RsCDPK members and the GTF file of “QZ-16”. To identify collinear blocks, MCScanX was employed with default parameters utilizing GTF gene annotation information [38].

### 4.3. Gene Structure and Conserved Motif Analysis

Phylogenetic relationships, gene structures, and conserved motifs were visualized using the TBtools software (v1.1041) [60]. The conserved motifs of RsCDPK proteins were analyzed using the Multiple Expectation Maximization for Motif Elicitation program (MEME, http://meme-suite.org/tools/meme, accessed on 26 August 2023). The maximum number of motifs was set to 15, with default parameters for the remaining settings. The amino acid sequences of all RsCDPK proteins were aligned using the Muscle method. The resulting alignment file was then used to construct a neighbor-joining tree in MEGA 7, with 1000 bootstrap values.

### 4.4. Phylogenetic and Motif Analyses of CDPK Genes

To evaluate the relationship between CDPK genes, evolutionary analysis was performed in radish, rice, and tomato. The AtCDPK genes from *A. thaliana* were downloaded from “The Arabidopsis Information Resources” database (https://www.arabidopsis.org/, accessed on 27 August 2023), whereas SlCDPK genes from tomato were downloaded from the “The Sol Genomics Network” database (https://solgenomics.net/, accessed on 28 August 2023). All CDPK sequences were aligned using the muscle method. The resulting alignment file was utilized to construct a neighbor-joining tree with 1000 bootstrap values using MEGA 7.

### 4.5. Plant Materials and Phenotypic Observation

HongYun and 505 are high-generation inbred radish lines showing crack-resistant and -sensitive root phenotypes, respectively, at the mature fleshy root stage. Radishes were planted in the greenhouse hydroponic experimental base of the Hubei Vegetable Research Institute using the nutrient solution method. A phenotypic investigation was performed at the mature stage of the radish taproot.

The fissure taproot cortices of HongYun and 505 were collected and fixed with an FAA fixative, as previously described [61]. Paraffin-embedded sections were prepared (Wuhan Servicebio Technology Co., Ltd., Wuhan, China) and observed using a microscope (Nikon Eclipse E100, Tokyo, Japan). 

### 4.6. Transcriptomic Sequencing

The fleshy taproot cortices were narrowly harvested from HongYun and 505 plants. Three biological replicates (three fleshy taproot cortices each) were used for the non-cracking pool (HongYun-1, HongYun-2, and HongYun-3), and three were used for the cracking pool (505-1, 505-2, and 505-3). The fleshy taproot cortices were frozen in liquid nitrogen and subsequently stored at −80 °C until further processing. Total RNA was extracted using the RNA Isolator Total RNA Extraction Reagent (Vazyme, Nanjing, China), following the manufacturer’s instructions. The quality of RNA was assessed by Nanodrop NC2000 (Thermo Scientific, Waltham, MA, USA), gel-electrophoretic apparatus DYY-6C (LIUYI, Beijing, China), and gel imaging system GelDoc 2000 (BIO-RAD, Hercules, CA, USA). The concentration and volume of each cDNA library were 25 ng/μL and 20 μL, respectively. Transcriptome sequencing was performed on a HiSeq X Ten platform (Illumina, San Diego, CA, USA). The Cutadapt 2.2 was used for data filtering to remove the 3′ end-band connector sequence and remove the average mass fraction below the Q20 Reads standard to obtain the clean data.

### 4.7. Identification of DEGs and Enrichment Analyses

The clean reads were aligned to the radish reference genome “QZ-16” using TopHat2′s upgraded HISAT 2.2.1. The expression levels were calculated by the fragments per kilobase of transcript per million mapped reads (FPKM). DEGs between HongYun and 505 were identified using the DESeq2 package [62,63]. The DESeq2 package offers statistical methods for identifying differential expression in digital gene expression data. The analysis process includes three main steps, namely normalization, dispersion estimation, and test for differential expression. Firstly, DESeq2 is used to read the counts file after comparison in R language, including sample classification information and grouping information. Then, the dds matrix data are constructed and normalized, and the results are obtained by the ‘results ( )’ function and assigned to the ‘res’ variable. The results of differential analysis were stored in ‘res’, which contained the main information such as gene ID, normalized gene expression value, fold change value after log_2_ transformation, and significant *p*-value. To control the false-discovery rate, the Benjamini–Hochberg procedure was applied to adjust the generated *p*-values. These methods are based on a model that utilizes the negative binomial distribution [64]. DEGs were considered significant if they had an adjusted *p*-value < 0.05, a fold change ≥2 in FPKM, and a false-discovery rate < 0.01.

To further analyze the DEGs, we employed the GOseq R package. This package utilizes the Wallenius non-central hypergeometric distribution and takes into account gene length bias in DEGs. GO enrichment analysis was performed to identify enriched functional categories associated with the DEGs [65]. Additionally, we conducted KEGG enrichment analysis using the KOBAS 3.0 [66].

### 4.8. RT-qPCR Validation

The transcriptional levels of HongYun and 505 were determined using RT-qPCR. The primer sequences for all genes are in Appendix A. Total RNA was extracted using RNA Isolator Total RNA Extraction Reagent (Vazyme), following the manufacturer’s instructions. The quality of RNA was evaluated by Nanodrop NC2000 (Thermo Scientific), gel-electrophoretic apparatus DYY-6C (LIUYI), and gel imaging system GelDoc 2000 (BIO-RAD) to ensure that the quality of the sample RNA was suitable for subsequent experiments. To synthesize cDNA, a RevertAid First Strand cDNA Synthesis Kit (Vazyme) was utilized, following the manufacturer’s protocol. RT-qPCR experiment was performed using ChamQ Universal SYBR qPCR Master Mix (Vazyme), following the manufacturer’s instructions. The standard conditions were as follows: initial denaturation at 95 °C for 3 min; followed by 39 cycles of 95 °C for 10 s, 58 °C for 10 s, and 72 °C for 15 s (fluorescence signal acquisition); followed by selecting the instrument default melting curve acquisition procedures. A CFX384 Touch Real-Time PCR Detection System (Bio-Rad Laboratories, Hercules, CA, USA) was used to analyze the transcription levels. The gene expression levels were calculated by the formula 2^−∆∆Cq^ method [67]. To normalize the RT-qPCR values and account for differences in total RNA levels, The radish *RNA polymerase-II transcription factor* (*Rs.RPII*) gene was used as the internal control [36]. All statistical analyses, such as those to calculate the row mean with the SD or SEM, were performed using the Prism software (GraphPad Software 8.0.2, La Jolla, CA, USA).

### 4.9. Vector Construction and Soybean Hairy Root Transformation

To generate the overexpression construct of RsCDPK21, the coding sequence of RsCDPK21 was amplified via PCR using Phanta Max Super-Fidelity DNA polymerase (Vazyme) from “505” and inserted into the binary vector pMDC83 directly under the control of the CaMV 35S promoter sequence using *BamH*I and *Kpn*I, which makes use of the ClonExpress II One Step Cloning Kit (Vazyme). The recombinant vector was introduced into *Agrobacterium rhizogenes* strain K599 (Weidibio, Anhui, China) and used to transform Williams 82 (a transgenic acceptor soybean cultivar). Soybean hairy root transformation was performed as previously described [68,69]. Root length was measured using a vernier caliper. *GmELF1b* was used as the internal control. All statistical analyses were carried out with the Prism software (GraphPad) like Row mean with SD or SEM. The data were analyzed with the LSD test, using SPSS software 18.0 (SPSS Inc., Chicago, IL, USA). Statistically significant differences (Student’s *t*-test) are indicated as follows: **, *p* < 0.01; ***, *p* < 0.001. GraphPad Prism software (GraphPad) was used for data processing and image visualization. The primer sequences for all genes can be found in Appendix A.

The transgenic roots of EV and RsCDPK21OE were collected and fixed with FAA fixative, following the previously described method [61]. Paraffin-embedded sections were prepared (Wuhan Servicebio Technology Co., Ltd.) and observed using a microscope (Nikon Eclipse E100).

## Figures and Tables

**Figure 1 ijms-24-15059-f001:**
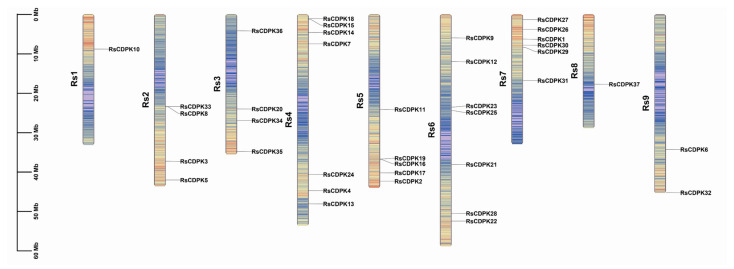
Chromosomal distribution of *RsCDPK* genes. The distribution of the 37 *RsCDPKs* is based on relative physical positions. Colors on the nine radish chromosomes indicate gene density (gene number/100 kb).

**Figure 2 ijms-24-15059-f002:**
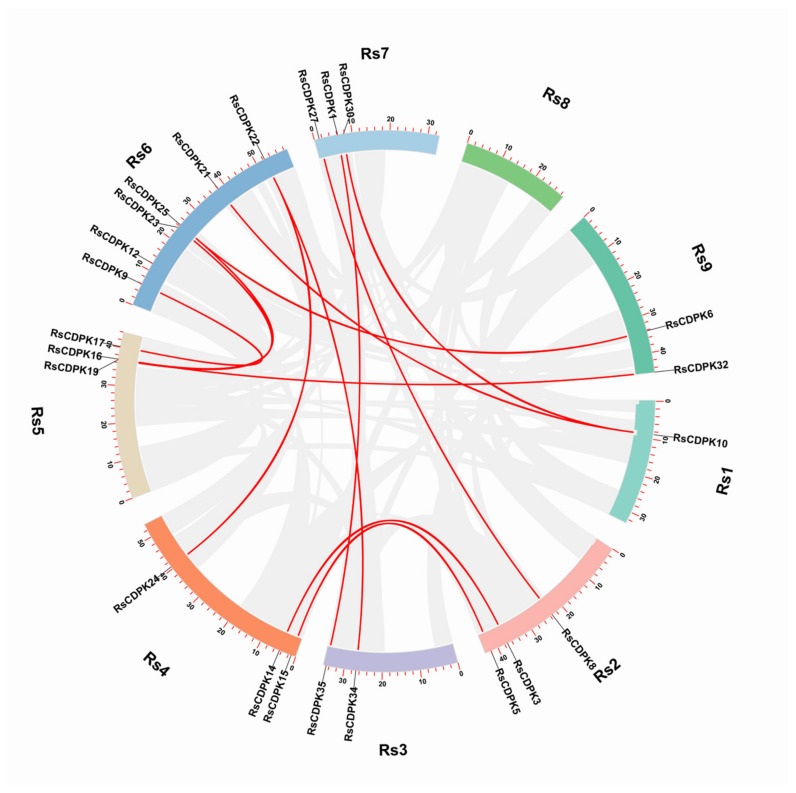
Genome collinearity analysis of *RsCDPK* genes. Gray lines depict all syntenic blocks in the nine chromosomes of radish. Red lines indicate the segmental duplication of *RsCDPK* genes.

**Figure 3 ijms-24-15059-f003:**
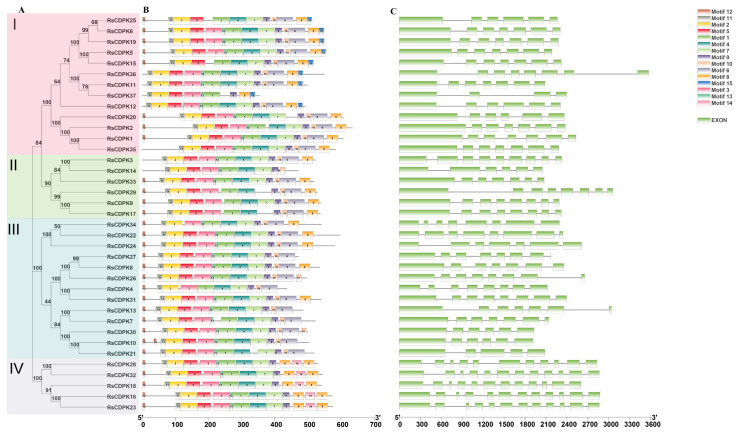
Phylogenetic tree, conserved motif distribution, and gene structure of *RsCDPK* genes. (**A**) The phylogenetic tree of RsCDPK proteins was constructed using MEGA 7 by the neighbor-joining method with 1000 bootstrap values and variable color backgrounds for different clades. (**B**) Conserved motif distributions of *RsCDPK* members. (**C**) Exons are represented by green boxes and introns by black lines. The size of the RsCDPK members is indicated at the bottom.

**Figure 4 ijms-24-15059-f004:**
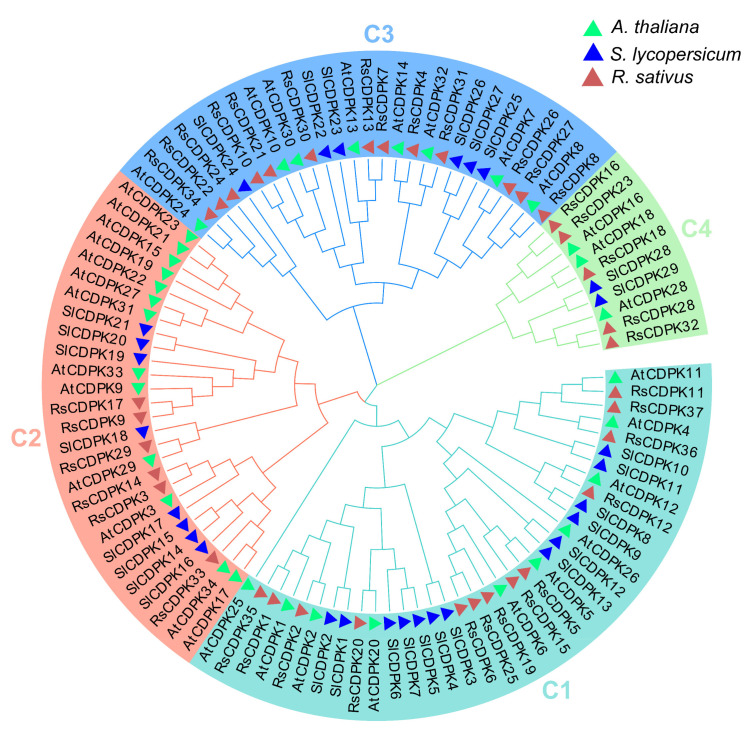
Unrooted phylogenetic tree of *RsCDPK* genes from *Arabidopsis*, radish, and tomato. Multiple sequence alignments of *RsCDPK* members were performed using the muscle method. The resulting aligned sequences were used to construct a phylogenetic tree in MEGA 7 using the neighbor-joining method with 1000 bootstrap replicates. The four subfamilies are highlighted in different colors. RsCDPK proteins are represented as follows: *Arabidopsis thaliana*, solid green triangles; *Raphanus sativus,* solid brown triangles; *Solanum lycopersicum,* solid blue triangles.

**Figure 5 ijms-24-15059-f005:**
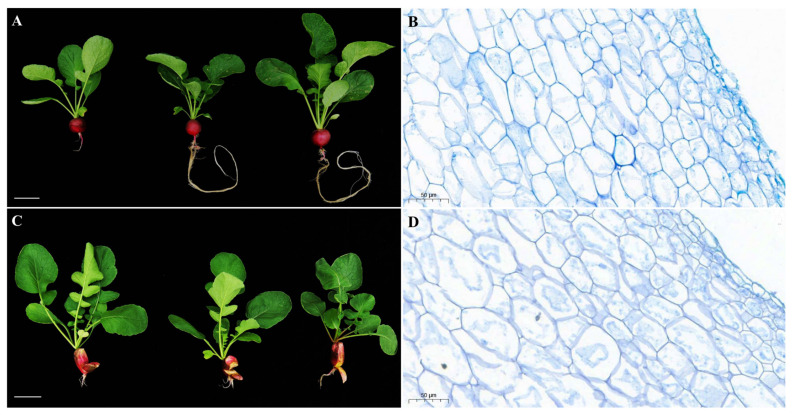
Phenotypic characterization of mature fleshy taproots in “HongYun” and “505”. Phenotype of (**A**) “HongYun” and (**C**) “505” plants. Scale bar = 5 cm. (**B**) Cytological observation of the cortex of “HongYun” and (**D**) “505”. Scale bar = 50 μm.

**Figure 6 ijms-24-15059-f006:**
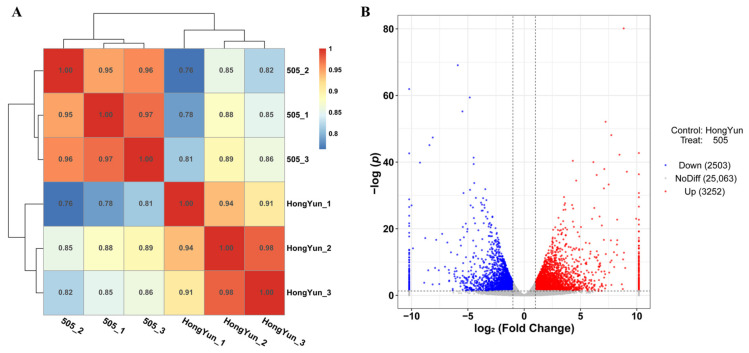
Analysis of Pearson correlation and differentially expressed genes in RNA-seq data. (**A**) Pearson correlation analysis of three non-cracking plant fleshy taproot cortices (HongYun_1, HongYun_2, and HongYun_3) and three cracking fleshy taproot cortices (505_1, 505_2, and 505_3). Pearson’s correlation coefficient values > 0.8 indicate that the correlation between samples is high. Based on the K-means method, the expression correlation between samples was clustered. *p*-value > 0.05. (**B**) Volcano map of differentially expressed genes. The abscissa is log_2_ (fold change), indicating the expression fold difference, and the ordinate is -log(*P*), indicating the significant gene expression results. The two vertical dashed lines in abscissa are two times the expression difference threshold; the horizontal dotted line in ordinate is *p*-value = 0.05 threshold. Red dots represent upregulated genes, blue dots represent downregulated genes, and gray dots represent non-significant differentially expressed genes.

**Figure 7 ijms-24-15059-f007:**
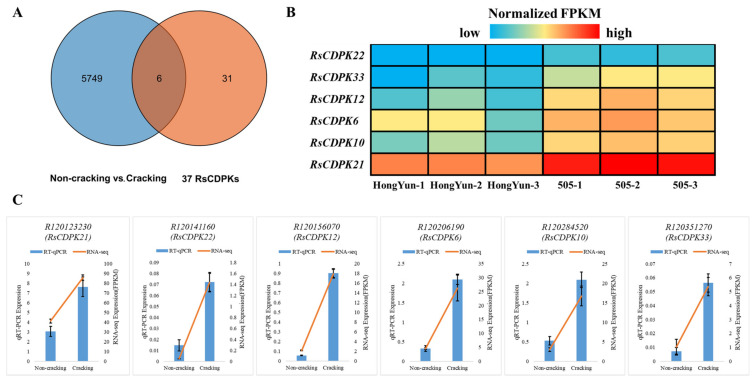
Venn diagram and expression level analysis of RsCDPK genes associated with radish taproot cortex. (**A**) Venn diagram of the 5755 differentially expressed genes (DEGs) and 37 RsCDPK genes. (**B**) Heatmap of fleshy taproot cortex-related RsCDPK DEGs. (**C**) RT-qPCR validation. The 2^−ΔΔ Cq^ method was used to calculate the relative expression. All values represent the means ± SD of three independent experiments (such as FPKM value of RNA-seq).

**Figure 8 ijms-24-15059-f008:**
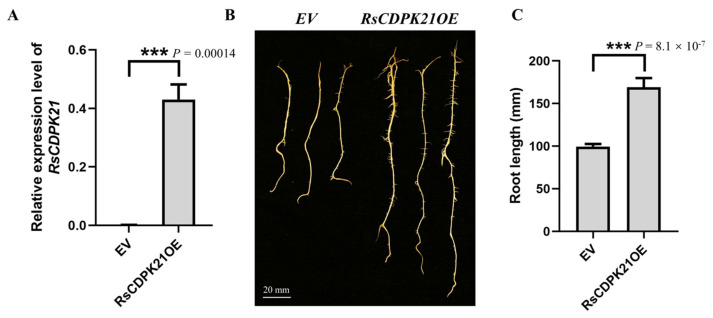
Transgenic root phenotype and root length statistics. (**A**) The relative expression levels determined by RT-qPCR of *RsCDPK21* in transgenic hairy roots constitutively express empty vector (EV) and *RsCDPK21OE* 14 days after rhizobial inoculation. The 2^−ΔΔ Cq^ method was used to calculate the relative expression. (**B**) Transgenic roots of EV and *RsCDPK21OE*. Scale bar = 20 mm. (**C**) Root length of the transgenic roots of EV and *RsCDPK21OE*. All values are the means ± SD of three independent experiments or 10 independent hair roots. Statistically significant differences (Student’s *t*-test) are indicated as follows: ***, *p* < 0.001.

## Data Availability

The data presented in this study are available upon request from the corresponding author.

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
