# Peer review of "Genome-Wide Identification of the CDPK Gene Family and Their Involvement in Taproot Cracking in Radish"

_ijms, 2023, doi:10.3390/ijms242015059_

Round 1
Reviewer 1 Report
Authors try to functionally characterise a group of RsCDPK genes. After phylogenetic and bioinformatic studies, Authors identified a group of six RsCDPK genes that are differentially expressed between cracking suseptible and cracking resistant radish (Rs) plants.
Authors indicated also that the overexpression of RsCDPK21 increases the length of transformed soybean hairy roots. However, the direct link between cracking and RsCDPK21 overexpression was not proved experimentally and here the conclusions are not supported by results.
Research increases the available information related to phylogenetics and functions of RsCDPKs genes in radish. References are appropriate.
Following comments should be addressed:
1. Figure 3B It is hard to read numbers of motifs. If the font size could not be increased try to build a Suppl. File containing information of motifs in each RsCDPK protein.
2. Fig 7 Try to increase the font size; the GO terms and KEEG pathways are not readable.
3. Section 2.8
Authors wrirte : „Notably, all of these genes were upregulated in the non cracking variety”. However, Fig. 8B shows the higher expression in Rs564 that is cracking-suscepible.
4. Section 2.9 Authors write that RsCDPK12 has the highest FPKM value- it is not true, Rather RsCDPK21 has the highest FPKM- according to Fig 8B. Maybe it is just the typographical error and in the text should be RsCDPK21 not RsCDPK12?
Correct the sentence: To verify the function of the six RsCDPK genes identified by Venn diagram analysis of the RsCDPK gene family members and transcriptome DEGs in regulating root development, the RsCDPK12 genes with the highest expression levels (FPKM values) were selected for further analysis.
To following: (red font-remove, blue font -add)
To verify the function of the six RsCDPK genes identified by Venn diagram analysis of the RsCDPK gene family members and transcriptome DEGs in regulating root development, the RsCDPK21 genes with the highest expression level (FPKM value) were was selected for further analysis.
5. Section 4.6
How the quality of RNA was assessed ?
Provide concentration and volume od cDNA libraries.
Provide name of software used to obtain clean reads.
Include the name of software used to align obtained seqences to reference genome.
6. Section 4.8
Provide method of RNA isolation. How the quality of RNA w3as assured?
Describe amount of RNA per sample.
Provide details of PCR steps.
Present the full name of Rs.RPII reference gene.
Include the citation of method used to calculate the relative gene expression.
Section 4.9
Provide name of used vector. How the cDNA was cloned into the vector? Provide for example names of restriction sites.
In the first sentence In Section 4.9 Authors write of RsCDPK12 and RsCDPK21. It should be only RsCDPK21.
Minor editing of English language required.
Reviewer 2 Report
Manuscript "Genome-wide identification of the CDPK gene family and their involvement in taproot cracking in radish" is very interesting.
General comments:
Authors identified 37 RsCDPK gene members from the long-read radish genome “QZ-16". Phylogenetic analysis revealed that the CDPK members in radish, tomato, and Arabidopsis clustered into four groups. Authors identified 13 segmental duplication events in the RsCDPK genes.
Detailed comments:
Figure 3A: You should state what measure was used to calculate similarity/difference. You should state what method was used to construct the phylogenetic trees.
Lines 157-159: Appropriate methods are used to identify relationships. The authors provided the software, unfortunately they did not provide any method.
Figure 4: You should state what measure was used to calculate similarity/difference.
Lines 176-177: "We observed no significant difference in the size 176 of the leaves or fleshy roots at the mature stage." Confirm by providing the results of the statistical analysis.
Lines 184-186: Confirm by providing the results of the statistical analysis.
Figure 6A: Provide a critical value for testing the significance of correlation coefficients.
Figure 6A: What method was used to construct the similarity dendrograms?
Figure 6A: The values of correlation coefficients should be written in a larger font.
Figure 6A: None of the correlation coefficients were statistically significant?
Figure 6B: "log2" - "2" should be in subscript.
Figure 6B: In the notation of the decimal logarithm, '10' is not given.
Figure 6B: "(p-values)": It is necessary to clarify exactly the difference of which values are involved!
Figure 7A: In the notation of the decimal logarithm, '10' is not given.
Figure 7A: "(p-values)": It is necessary to clarify exactly the difference of which values are involved!
Figure 8C: Complete with LSD values for comparing average values. The method used should be described in the methodology section.
Figure 9A: Complete with LSD values for comparing average values. The method used should be described in the methodology section.
Figure 9C: Complete with LSD values for comparing average values. The method used should be described in the methodology section.
Lines 396-397: "A phylogenetic tree of RsCDPK was 396 constructed using MEGA 7.". This is an administration of the program, not a description of the methodology. You should state what measure was used to calculate similarity/difference. You should state what method was used to construct the phylogenetic trees.
Line 426: "DEGs between Rs379 and Rs564 were identified using the DESeq R package (1.10.1).". The significance of differences can be identified by an appropriate test, not by a package. The description of the statistical methodology should be completed by stating the relevant statistical methods used in the implementation of the experiment. Providing only the name of the program shows a machinic (uninformed) approach to the reliability of scientific research.
Paper needs major revision.
Reviewer 3 Report
The manuscript is very well written and holds immense significance for understanding the molecular basis of taproot cracking in radish.
The comments to be addressed include:
Line 14- Change "its functions in radish taproot cracking remain" to "its function in radish taproot cracking remains"
Expand "EF" used at multiple places in Introduction section.
Line 282 and 291 do not agree as at Line 282 RsCDPK12OE is mentioned whereas at Line 291 RsCDPK21OE. Kindly correct it.
Minor editing in English language required
Round 2
Reviewer 1 Report
Authors addressed all issues presented in the review nad significantly improved the manuscrit. I have no comments.
Author Response
Thanks for your kind advice and detailed suggestions. Those comments are all valuable and very helpful for revising and improving our paper, as well as the important guiding significance to our research.
Reviewer 2 Report
10. Figure 6B: "log2" - "2" should be in subscript.
Response 10:
Thank you for bringing this to our attention. We have made the correction in the revised manuscript.
Ad. 10. Figure 6B: What does "lg" mean? Still!
Figure 6B: "p-values"?
12. Figure 6B: "(p-values)": It is necessary to clarify exactly the difference of which values are involved!
Response 12:
Thanks for your comments. We have made the correction in the revised manuscript.
Ad. 12. Still: "(p-values)". On the ordinate axis.
15. Figure 8C: Complete with LSD values for comparing average values. The method used should be described in the methodology section.
Response 15:
Thank you very much for carefully reviewing our manuscript. We have modified these descriptions in the revised manuscript section 4.8 according to your suggestion. The original Figure 8C was changed to the current Figure 7C.
Ad 15. LSD values are still missing from the individual figures.
Part of the caption under Figure 7 is redundant: "All statistical analyses were performed using the Prism software (GraphPad 8.0.2)." Specify the methods used, not the program. A scientific article must be written in such a way that anyone can repeat it. Pointing to the software itself is not an informative message.
17. Figure 9C: Complete with LSD values for comparing average values. The method used should be described in the methodology section.
Response 17:
Thank you very much for carefully reviewing our manuscript. We have modified these descriptions in the revised manuscript section 4.9 according to your suggestion. The original Figure 9C was changed to the current Figure 8C.
Ad. 17. LSD values are still missing from the Figure.
19. Line 426: "DEGs between Rs379 and Rs564 were identified using the DESeq R package (1.10.1).". The significance of differences can be identified by an appropriate test, not by a package. The description of the statistical methodology should be completed by stating the relevant statistical methods used in the implementation of the experiment. Providing only the name of the program shows a machinic (uninformed) approach to the reliability of scientific research.
Response 19:
Thank you for the insightful comments and suggestions. We have made the modifications section 4.7 and cited the related references (as follow) in the revised manuscript according to your suggestion.
[62] Wang, L.; Feng, Z.; Wang, X.; Wang, X.; Zhang, X., DEGseq: an R package for identifying differentially expressed genes from RNA-seq data. Bioinformatics 2010, 26, (1), 136-8.
Ad. 19. The authors wrote that they made modifications to Section 4.7. Unfortunately, these are not substantive modifications. "Clarification" according to the Authors was to provide only the name of the package. The methodology should list all the methods used during the study: statistical too. It is imperative that the manuscript be improved so that anyone can repeat the experiment and the analyses of the data obtained. Without this, the manuscript is not suitable for publication in any scientific journal.
Paper need major revision.
Round 3
Reviewer 2 Report
The manuscript still needs improvement. As I mentioned earlier, the authors machinely use packages by "selecting" results from them without any basis for the validity of using specific methods. Moreover, they choose the wrong methods. This is the third round of reviews, and the errors I pointed out in the first round still remain uncorrected.
12. Figure 6B: The notation "(p)" instead of "(p-value)" will suffice.
15. I have no reservations about the LSD test. However, I note once again that no specific LSD values are given for comparing individual pairs. This should be supplemented.
17. I have no objection to the p-values given. However, I note once again that no specific LSD values are given for comparing individual pairs. This should be supplemented.
In the attached scans, the authors presented only a partial post-hoc analysis for Fisher's method. When testing the difference between two averages, there is no need for multiple comparisons (there is only one comparison) and a simple Student's t-test is sufficient. This should be improved.
"The DESeq2 package offers statistical methods for identifying differential expression in digital gene expression data."
What methods?
"
Firstly, DESeq2 is used to read the counts file after comparison in R language, including sample classification information and grouping information
."
Based on which classification method?
"Then, the dds matrix data is constructed and normalized, and the results are obtained by the 'results ( )' function and assigned to the 'res' variable."
What is a dds matrix? How is it constructed?
What results are obtained?
What is the "results" function? Please under the mathematical formula.
Why was the transformation made? The authors do not mention anything about studying the distribution.
Round 4
Reviewer 2 Report
The authors revised the manuscript according to all comments and suggestions. It can be published in this form.